# Explaining Plan Quality Differences

**Primary Keywords:** *(5) Human-aware Planning and Scheduling*

## Abstract

In this paper we describe a method for explaining the differences between the quality of plans produced for similar planning problems. The method exploits a process of abstracting away details of the planning problems until the difference in the quality of the solutions they support has been minimised. We give a general definition of a valid abstraction of a planning problem. We then give the details of the implementation of a number of useful abstractions. Finally, we present a depth-bounded breadth-first search algorithm for finding suitable abstractions for explanations; and detail the results of an evaluation of the approach.

## 1 Introduction

In mixed-initiative planning settings, human and automated planners interact and collaborate to produce satisfactory plans. An automated planner is used to produce plans quickly and a human can then add constraints and preferences to the model until they are satisfied with the resulting plan produced by the automated planner. After each addition of a constraint the newly generated plan can be of better or worse quality compared with the version generated without the constraint, or the problem could even become unsolvable. In each of these cases it would be useful to have accompanying explanations for why additional constraints lead to a difference in plan quality.

The setting in which these types of explanations are useful does not have to be a mixed-initiative setting. We assume that the setting is as follows: there is a planning model, $\Pi$, a constraint, $c$, which when applied to $\Pi$ limits it's possible plans, a solution plan for $\Pi$, $\pi$, and a solution plan for $\Pi + c$, $\pi'$, where there is a difference in the quality of $\pi$ and $\pi'$. A special case for this is where $\pi'$ does not exist, that is the model $\Pi + c$ is unsolvable. As in Krarup *et al.* 2021, we refer to this process as *model restriction*. We restrict planning models such that they admit only solutions that obey a certain constraint. The problem is then to explain why there is a difference in the quality of $\pi$ and $\pi'$. We assume that an explanation of the form "the difference is because of the constraint $c$" is not helpful.

While there is a long history of work on explanation in AI, most work on explanation of plans (XAIP) is relatively recent. Fox et al. 2017 highlight contrastive 'why' questions as being important for plan explanation, and describe a number of different types of these questions and possible responses. Chakraborti et al. 2017 adopt the position that explanation is a model reconciliation problem – namely, that the need for explanation is due to differences between the agent's and the human's model of the planning problem. Eifler et al. 2020 approach answering local contrastive questions by explaining the reason that a contrast case $B$ was not in the plan, or a feature of the plan, by using the properties that would hold if $B$ *were* the case. Kasenberg et al. 2019 focus on justifying an agent's behaviour based on deterministic Markov decision problems.

We are not aware of any work that focusses on explaining the difference in the quality of plans. However, there is work on explaining why planning models are unsolvable. Gobeldecker et al. 2010 focus on finding changes to the initial state that would make the planning problem solvable, and provide an algorithm to produce these excuses in a reasonable time. Sreedharan *et al.* use abstractions to explain the unsolvability of planning problems, as opposed to explaining the difference in quality of problems. They do this by considering relaxations of the planning problem until a solution can be found, and then looking for landmarks of this relaxed problem that cannot be satisfied in less relaxed versions of the problem. They assume that the existence of certain predicates cause a planning problem to be unsolvable. Thus the only abstraction they use is the removal of predicates from the model. We recognise a larger set of useful abstractions for AI planning systems, especially those that have languages to specify temporal characteristics of plans.

An explanation for why two similar problems produce different quality plans should focus on the essential characteristics of the problems. For example, in a delivery problem, a user might ask why a particular truck was used rather than an alternative. The answer might be that the selected truck is better because of a weight limit on a bridge, or refrigeration properties of the truck. Inspired by Lipton's difference condition(Lipton 1990) we want to find and explain more accurately the causal differences between the original and hypothetical plans. Finding this kind of explanation requires abstracting away unimportant details like drivers, cargo, and perhaps route details. We show how these explanations can be generated by abstracting features of the planning problem until the two plans become *equi-quality* – that is, of equal or similar quality. We can then explain why one plan is better or

worse than the other in terms of the abstracted features that impacted the difference in plan quality between the two.

The primary contribution of this paper is to define a framework for the use of abstractions to explain plan quality differences. In this paper, we first define a valid abstraction for a planning model, these can be used to extend this work for new abstractions. We then introduce a running example we will use to motivate some useful abstractions. We formalise the implementation of a number of useful abstractions for explaining quality differences of temporal plans. We show how these abstractions fit our definition of an abstraction. We detail a proof-of-concept breadth-first search algorithm for finding suitable abstractions for explanation, and present the results of an experiment evaluating it.

## 2 Planning Formalism

**Definition 1.** *A **planning model** is a pair $\Pi = \langle D, Prob \rangle$. The domain $D = \langle Ps, Vs, As, arity \rangle$ is a tuple where $Ps$ is a finite set of predicate symbols, $Vs$ is a finite set of function symbols, $As$ is a set of action schemas, called operators, and $arity$ is a function mapping all of these symbols to their respective arity. The problem $Prob = \langle Os, I, G, M, T \rangle$ is a tuple where $Os$ is the set of objects in the planning instance, $I$ is the initial state, $G$ is the goal condition, $M$ is a plan-metric function from plans to real values (plan costs) and $T$ is a set of timed initial literals.*

A set of atomic *propositions* $P$ is formed by applying the predicate symbols $Ps$ to the objects $Os$ (respecting arities). One proposition $p$ is formed by applying an ordered set of objects $o \subseteq Os$ to one predicate $ps$, respecting its arity. This process is called "grounding" and is denoted with $ground(ps, \chi) = p$, where $\chi \subseteq Os$ is an ordered set of objects. The inverse of this function $ground^{-1}(p) = \langle ps, \chi \rangle$ returns the predicate symbol and objects. Similarly the set of *ground functions* $V$ are formed by applying the function symbols $Vs$ to $Os$.

A state $s$ consists of a time $t \in \mathbb{R}$, a logical part $s_l \subseteq P$, and a numeric part $s_v$ that describes the values for the ground functions at that state. The initial state $I$ is the state at time $t = 0$. We use the function $S(\Pi)$ to denote the state space for a model $\Pi$, i.e. all states reachable from the initial state in $\Pi$. The goal $G = g_1, ..., g_n$ is a set of propositions, including a subset of the logical state variables, $P$, and a set of constraints over numeric state variables in $V$, that must hold at the end of an action sequence for a plan to be valid.

Similarly to propositions and functions, the set of ground actions $A$ is generated by the substitution of objects for operator parameters. Each ground action is defined as follows:

**Definition 2.** *A **ground action** $a \in A$ has a duration $Dur(a) = \langle lb, ub \rangle$ which constrains the length of time that must pass between the start and end of $a$; a start (end) condition $Pre_{\vdash}(a)$ ($Pre_{\dashv}(a)$) which must hold in the state that $a$ starts (ends); a numeric condition $Pre_{\leftrightarrow}(a)$ which must hold throughout the entire execution of $a$; add effects $Eff(a)_{\vdash}^{+}, Eff(a)_{\dashv}^{+} \subseteq P$ that are made true at the start and end of the action respectively; delete effects $Eff(a)_{\vdash}^{-}, Eff(a)_{\dashv}^{-} \subseteq P$ that are made false at the start and end of the action respectively; and numeric effects $Eff(a)_{\vdash}^{n}$,*
*$Eff(a)_{\leftrightarrow}^{n}$, $Eff(a)_{\dashv}^{n}$ that act upon some $v \in V$, at the start of an action, continuously over the entire execution of an action, and the end of an action.*

For ease of notation we allow access to multiple types of effects or preconditions through the ground action functions at once. For example for some ground action $a$, $Eff^{+}$ denotes all add effects of $a$, $Pre_{\vdash\dashv}(a)$ denotes all start and end preconditions of $a$ but not invariant conditions, $Eff(a)$ denotes all effects of $a$ including numeric effects, etc.

A plan is a sequence of grounded actions, $\pi = \langle a_1, a_2, \ldots, a_n \rangle$ each with a respective time denoted by $Dispatch(a_i)$ and duration $Dur(a_i)$. A valid plan is a plan that transforms the initial state from $I$ into a state $s$ such that $s \models G$; such that the start and end preconditions of all actions are satisifed at the time they start/end; all invariant conditions hold throughout the durations of each action, and all actions execute for their respective durations.

The plan-metric function is, by default, the makespan of the plan. Therefore usually, and throughout this paper, the higher the metric of the plan the worse the quality of the plan. More generally, the metric assesses plan quality by taking into account both the extent to which a plan respects user preferences and also the costs associated with the choices of action or combinations of actions within a plan. It is often the case that plans fail to meet expectations because of a mismatch in the way that plans are evaluated.

Timed initial literals (TILs) were introduced as part of PDDL2.2 (Hoffmann and Edelkamp 2005), allowing an initial state to include simple effects triggered at specified times, regardless of the plan. Each timed initial literal $t \in T$ is a tuple $t = \langle t_{time}, t_v \rangle$ where $t_v$ is a proposition which becomes true or false, or a numeric effect which acts upon some $n \in V$ at the time $t_{time}$.

## 3 Food Delivery Problem

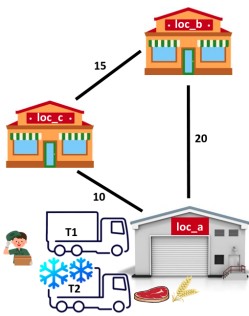

Figure 1: A diagram of the food delivery domain.

As a reference example, we use a simplified version of a food delivery domain. There are two trucks: truck T1 is a normal truck, whereas truck T2 is refrigerated. There is only one driver. The goal of the problem is to deliver the meat and cereals packages to their respective locations. The meat must be delivered to loc_b and the cereals to loc_c. Both trucks, the meat, and the cereals are initially at loc_a. It takes 20 minutes for a truck to move between the loc_a and loc_b,

15 minutes from loc_b and loc_c, and 10 minutes between loc_a and loc_c. The meat will spoil after 22 minutes, unless it is refrigerated. The cereals do not spoil.

For simplicity, we define two possible routes. Route 1 is any plan in which we load the meat and cereals packages into any truck, drive from loc_a to loc_b and deliver the meat, and finally drive from loc_b to loc_c and deliver the cereals. Route 1 takes 35 minutes. Route 2 is any plan in which we load the meat and cereals packages into any truck, drive from loc_a to loc_c and deliver the cereals, and finally drive from loc_c to loc_b and deliver the meat. Route 2 takes 25 minutes. Note, not every route is necessarily possible with every truck. The optimal plan is to perform Route 2 with the refrigerated truck T2. We refer to this plan as $\pi^*$ throughout.

## 4   Abstraction

In the setting of model restriction described in Section 1, we want to explain why there is a difference in the quality of two plans $\pi$ and $\pi'$ based on a constraint $c$ added to a planning model $\Pi$. We propose to do this through abstracting away elements of one of the planning models until $M(\pi) = M(\pi')$. We will abstract away details from $\Pi$ if and only if $M(\pi) > M(\pi')$, and we will abstract away detail from $\Pi+c$ if and only if $M(\pi') > M(\pi)$. An abstraction of a planning model can be thought of as a relaxation of the problem that the planning model describes. For any planning model $\Pi'$ to be an abstraction of $\Pi$ then every solution of $\Pi$ must be a solution of $\Pi'$. Therefore, we ensure that by abstracting away details from the model with the lesser quality plan that we are always preserving valid plans for this model and therefore making the problem as easy to solve, and likely easier.

There is a large body of work on abstraction in planning. The majority of this work focuses on the use of abstractions for computing heuristics. A planning problem can be abstracted, making it easier to solve, this abstracted solution can then be used as an estimate for the actual solution. One abstraction for computing a heuristic is the relaxed planning graph heuristic (Hoffmann and Nebel 2001), where all delete effects are removed from the planning model to make the problem easier to solve. This is then used to generate a relaxed plan that provides an estimate of distance to a goal state in the original problem. Another example of an abstraction for computing heuristics is merge-and-shrink heuristics (Helmert et al. 2014), where a planning model is separated into transition graphs on each proposition. These transition graphs can then be merged and shrunk until a good estimate for the total transition system implied by the original planning model is found that can then be used for generating a heuristic. These are both examples of abstractions under our definition. Helmert et al. 2007 define what a general abstraction is in the context of AI planning based on a labeled state transition system that maps to the semantics of a planning system. They do this for providing flexible abstraction heuristics.

In this section we propose a more general approach to abstraction, where state merging is one realisation, based on the idea that abstraction is a relaxation. We define the space of legal abstractions using a labeled state transition system (LST), which represents the state space corresponding to the grounding of a planning problem, and a simple temporal network (STN), which represents the temporal constraints and orderings on possible solutions of a planning problem. We rigorously define what a valid abstraction of a planning model is as we later give the implementations of model surgeries that abstract planning models. We use the definitions in this section to prove that these model surgeries are valid abstractions. Abstractions that are not valid could make planning problems harder to solve, and so would not be useful for providing explanations.

The states of a planning model and transitions between them via applying actions can be modeled with an LST.

**Definition 3.** *A labelled state transition system (LST), $\tau$, is a triplet $\langle S, L, T \rangle$, where $S$ is a set of states, $L$ is a set of labels and $T \subseteq S \times L \times S$ is a set of transitions. A path, $\pi$, in $\tau$ is a pair in $S \times L^*$, consisting of a state and a finite sequence of labels $l_0, ..., l_{n-1}$, such that there is a sequence of states, $s_0, ..., s_n$ such that $s=s_0$ and, for each $i=0, ..., n\text{-}1$, $(s_i, l_i, s_{i+1}) \in T$.*

Given an LST $\tau$ we define an abstraction, $\tau'$, as follows:

**Definition 4.** *An LST, $\tau' = (S', L, T')$, is an* abstraction *of LST, $\tau = (S, L, T)$, with respect to initial state, $I \in S$, and goal set, $G \subseteq S$, if there is a mapping, $f : S \rightarrow S'$, such that for every transition, $\langle x, l, y \rangle \in T$, from a reachable and relevant state, $x \in S$, there is $\langle f(x), l, f(y) \rangle \in T'$.*

An LST can be derived from a planning model:

**Definition 5.** *Let $\Pi$ be a planning model; the LST derived from $\Pi$ is $\tau = \langle S, L, T \rangle$. $S$ is the set of all valuations of valid groundings of $Ps$ and $Vs$. $L$ is the set of labels corresponding to the ground actions $A$. $T = \{(s, o, s') | s \in S, \text{the ground action } o \text{ represents is applicable in} s, s' \text{ is the resulting state of applying the ground action } o \text{ represents in } s\}$. We define the function $\sigma(\Pi) = \tau$ to denote the derivation of an LST from a planning model $\Pi$.*

In planning problems, only states that can be traversed to from the initial state are *reachable* and only states that support paths to a goal state are *relevant* to the solution:

**Definition 6.** *Given an LST, $\tau = (S, L, T)$, a* problem *is a pair, $(I, G)$, such that $I \in S$ and $G \subseteq S$, and these are referred to as the initial state and goal set of the problem.*

**Definition 7.** *Given an LST, $\tau = (S, L, T)$, and a* problem, *$(I, G)$, a state, $s \in S$, is* reachable *if there is a path from $I$ to $s$ in $\tau$. A state, $s \in S$, is* relevant *if there is a path in from $s$ to some state $g \in G$ in $\tau$.*

Instead of thinking about abstractions of boundless LSTs, for planning problems we can think of abstractions of LSTs with relevant problems. Definition 4 can be generalised to define an abstraction that operates over all problems in some family of problems, $\mathcal{P}$, for a given LST.

**Definition 8.** *An LST, $\tau' = (S', L, T')$, is a* general abstraction *of LST, $\tau = (S, L, T)$, with respect to a set of problems, $\mathcal{P}$, if $\tau'$ is an abstraction of $\tau$ for every problem, $(I, G) \in \mathcal{P}$.*

The next set of corollaries trivially follow from the set of definitions above. These are useful for our presented proofs

in Section 5 and for proving some properties about abstractions that are generally useful, and expected from the definition of an abstraction.

Corollary 1 is very informative for abstractions for planning problems. Many relaxations of a planning problem will add transitions to the LST of the planning problem. Through this corollary, we can know that these suite of relaxations are all valid abstractions. Note that not all relaxations to a planning problem are trivially valid abstractions due to this abstraction. For example, if we remove negative effects from a planning problem but we allow negative preconditions, then this will remove edges in the derived LST. This is, of course, not the case if modelling language does not allow negative preconditions where this relaxation would be a valid abstraction trivially proved by this corollary.

Corollary 2 is a more formal way of denoting this effect removal abstraction that is common in relaxations of planning problems.

Corollary 3 is similarly informative. As abstractions of planning problems are relevant to a problem, $(I, G)$, then any relaxation of a planning problem that affects only states that are not relevant to the solution of the problem, of which there are often many, is a valid abstraction shown by this corollary.

Finally, Corollary 4 shows that state merging is valid according to our more general definition of an abstraction. This is useful as state merging abstractions have been long used as powerful tools in planning. However, we do not rely on this corollary for any of our proofs.

**Corollary 1.** *A second specific type of abstraction that falls trivially within those defined in Definition 4 is edge insertion abstraction: the addition of transitions to an LST, with $f$ as the identity function, certainly contains all transitions in $T$.*

**Corollary 2.** *For a planning model $\Pi$ with an initial state $I$, goal state $G$, and ground actions $A$ we can remove an effect $e \in Eff^{+-}_{\vdash\dashv}(a) \cup Eff^{n}_{\vdash\leftrightarrow\dashv}(a)$ for some action $a \in A$, to produce a planning model $\Pi'$. The LST $\tau'$ with the problem $(I, G)$ derived from $\Pi'$ is an abstraction of the LST $\tau$ with the problem $(I, G)$ derived from $\Pi$, if $e$ does not make any reachable and relevant condition $c \in Pre_{\vdash\leftrightarrow\dashv}(a')$ true for any $a' \in A$ or satisfy any goal $g \in G$.*

**Corollary 3.** *Removing states $s$ and transitions $\langle s, l, y \rangle$ where $s$ is not reachable or relevant for any problem $\mathcal{P}$ is a legal abstraction.*

**Corollary 4.** *The state merging abstraction is an abstraction according to Definition 4 using the following construction: states $u, v \in S$ are merged into the new state, $m$, by taking $S' = S \setminus \{u, v\} \cup \{m\}$ and $f(s) = s$ if $s \notin \{u, v\}$, $f(u) = f(v) = m$, and $T' = \{\langle f(s), l, f(s') \rangle | \langle s, l, s' \rangle \in T\}$. This is an abstraction by construction.*

To model plans in which action durations, temporal happenings, and temporal constraints matter; we can create a Simple Temporal Network (STN) (Dechter, Meiri, and Pearl 1991) on top of the LST from Definition 5 that dictates the timings of the state transitions in the LST. An STN is a graph whose vertices represent time points and weighted edges represent the maximum/minimum separation between these.

**Definition 9.** *A simple temporal network (STN), $G$, is a directed graph denoted by the triplet $\langle V, E, L \rangle$, where $V$ is a set of vertices, $E \subseteq V \times V$ is a set of edges, and $L \subseteq E \times \mathbb{R}$ is a set of labels applied to edges to represent temporal separation between vertices.*

An STN is constructed over an LST, $\tau$, as follows: given a finite sequence of labels in $\tau$, $\{a_1, a_2, ..., a_n\}$. the corresponding STN has $n + 2$ vertices, labelled $I$, $a_i$ (for $i = 1...n$) and $G$. The edges of the STN comprise an edge from $a_i$ to $I$ and from $a_{i+1}$ to $a_i$, each weighted $-\epsilon$ (where $\epsilon$ separates interfering actions), an edge from $G$ to $a_n$ weighted $0$, and, for each pair of labels, $a_i$ and $a_{i+j}$, that represent the start and the end of the same durative action instance, edges from $a_{i+j}$ to $a_i$ weighted with $-Dur(a_i)[0]$ and from $a_i$ to $a_{i+j}$ weighted $Dur(a_i)[1]$, i.e. for the edge $e_1$ from $a_i$ to $a_{i+j}$, $l(e_1) = Dur(a_i)[1]$, and for the edge $e_2$ from $a_{i+j}$ to $a_i$, $l(e_2) = -Dur(a_i)[0]$. If any action starts or ends are not paired off, the label sequence is not a valid plan; the sequence of labels is otherwise a valid plan if and only if it is both a valid path in the LST and also a consistent STN.

TILs can be captured within the framework of the LST and STN described above as follows. Given a TIL, $t = \langle t_{time}, t_v \rangle$, specifying effect $t_v$ occurs at time $t_{time}$, to be added to a planning model $\Pi$, a new proposition, $t$, is created and added to the initial state, an action, $TIL_t$ is created with precondition $t$, that deletes $t$ and with add effects $t_v$ and $doneT$; $doneT$ is added to the goal. An LST is then created in the usual way. The STN created for this temporal domain is then adjusted by adding edges of weight $t_{time}$ from $I$ to $TIL_t$ and $-t_{time}$ in the opposite direction. In this model, a valid plan will be forced to contain exactly one copy of the action $TIL_t$, in order to satisfy the goal. No additional copies can appear because of the deleted precondition. The temporal constraints can only be satisfied if $TIL_t$ occurs at exactly time $t_{time}$, along with all the other constraints of the temporal structure of the plan.

We can then define an abstraction of an STN:

**Definition 10.** *An STN $G' = \langle V', E', L' \rangle$ is an abstraction of STN $G = \langle V, E, L \rangle$, where $V' \subseteq V$ and $E' \subseteq E$, and if for any two vertices $v, v' \in V$ that are connected by an edge $e \in E$ where there is a label $l$ for $e$ in $L$, we have $l' \in L'$ where $l'(e) \leq l(e)$, if $v, v' \in V'$ and $e \in E'$.*

Finally, as we are working with temporal-numeric planning domains, we can define what a valid abstraction of a planning model is based on their derived LST and STN:

**Definition 11.** *A planning model $\Pi'$ is an abstraction of $\Pi$ if the LST $\tau'$ and the STN $G'$ derived from $\Pi'$ are abstractions of the LST $\tau$ and STN $G$ derived from $\Pi$.*

In a perfect world, we would be able to explain why a model is not solvable, by adding states, labels, and transitions to the LST of a planning model. The LST would have to represent the entire state space of a problem. LSTs can be large even for simple planning problems (Helmert 2009).

Not only can it become infeasible to explicitly represent the LSTs for these problems, but it can become intractable to realise what modifications to the LST we must make for precise abstractions. We instead make changes to the lifted representation of the planning problem with model surgeries.

This can simplify the abstraction process. However, it leads to abstractions having potentially larger affects to the state space than expected.

## 5 Abstraction Implementation

In this section we describe the implementation of abstractions that we have identified as important for explanation. We motivate each abstractions use in explanation with an example, we give the formal process of performing the abstraction on a planning model, and show how these fit our definition of abstraction. We prove proposition 1 for each abstraction $\alpha$. This proposition must hold to guarantee that the model surgeries we propose do not reduce the number of valid plans for a model, and therefore can be used to search for valid solutions or solutions of better quality.

**Proposition 1** (Valid Abstraction). *Given an LST $\tau = \langle S, L, T \rangle$ and STN $G = \langle V, E, L \rangle$ both derived from a planning model $\Pi$ and an LST $\tau' = \langle S, L', T' \rangle$ and STN $G = \langle V', E', L' \rangle$ derived from the planning model $\Pi_\alpha$, $\tau'$ is a valid abstraction of $\tau$ and $G'$ is a valid abstraction of $G$, and therefore $\Pi_\alpha$ is a valid abstraction of $\Pi$.*

Each proof of a valid abstraction in this section follows the same approach. Definition 4 defines what it means for a planning problem to be an abstraction of another. Each proof consists of showing that for $\tau$ $\tau'$, $G$, and $G'$ derived from $\Pi$ and $\alpha(\Pi)$, $\tau'$ is an abstraction of $\tau$ and $G'$ is an abstraction of $G$ and therefore $\alpha(\Pi)$ is an abstraction of $\Pi$.

For ease of notation, we define two new functions $Prop : Ps \rightarrow P$ and $Pred : P \rightarrow Ps$. The function $Prop$ takes a set of predicates and returns each of the propositions in $P$ that were formed from it's grounding, i.e. for a set of predicates $ps$, $Prop(ps) = \{p \in P | \exists ps' \in ps, \chi \subseteq Os : ground^{-1}(p) = \langle ps', \chi \rangle\}$. The function $Pred$ takes a set of propositions and returns the the set of predicates that were grounded for them to be formed, i.e. for a set of propositions $p$, $Pred(p) = \{ps \in Ps | \exists \chi \subseteq Os : ground(ps, \chi) = p\}$

### 5.1 Abstracting Predicates

Given the example spoken about in Section 3 with the plan being presented to a user as one in which truck T2 performs route 2, the user might instead prefer truck T1 to be used. Through the system of model restriction we can force the planner to ensure that the truck T1 is used throughout the plan. The resultant plan will consist of truck T1 performing route 1. This is because truck T1 cannot perform route 2 as the meat will spoil at time 22 and it is not possible to extend the life of the meat in the unrefrigerated truck T1. Route 2 takes only 25 minutes. However more than 22 minutes would have passed before the meat was delivered and so it would no longer be fresh. This new plan $\pi$ takes 35 minutes compared to the original 25 minutes.

Through the use of abstractions we can determine the cause of the disparity in solution quality. If we abstract away the predicates, $ps \in Ps$, that are responsible for modelling the need for refrigeration, then we can produce a plan, $\pi'$, in which truck T1 can perform route 2. This is a plan of the same quality as the original plan, but we are instead using truck T1 rather than truck T2 as the user expected. We can

therefore produce an explanation such as "if there were no need for refrigeration, then truck T1 could be used such as in $\pi'$, otherwise our new plan will be $\pi$ which is slower by 10 minutes". Abstracting a predicate consists of removing the predicate from everywhere that is appears in the model.

The formal process for abstracting a predicate from a planning model is as follows. Given a planning model $\Pi$ and a set of predicates $ps \subseteq Ps$ the abstracted model is $\Pi_{ps} = \langle D, Prob \rangle$, where $D = \langle Ps \setminus \{ps\}, Vs, As', arity \rangle$ and $As' = \{a' | \forall a \in As : Dur(a') = Dur(a), Pre(a') = Pre(a) \setminus \{ps\}, Eff(a') = Eff(a) \setminus \{ps\}\}$; and $Prob = \langle Os, I \setminus \{p\}, G \setminus \{p\}, M, T \rangle$, where $Prop(ps) = p$.

*Proof.* Removing a predicate $ps$ from the planning model is a valid abstraction. Removing a predicate obviously causes all preconditions involving that predicate to be removed. However, it also removes all effects (positive and negative) on that predicate, but these effects no longer matter, since no transition depends on them. This operation is equivalent to merging states that are otherwise identical except for the presence or absence of $ps$.

This can be seen by realising that, for the LST $\tau$ with a set of problems $\mathcal{P}$, the mapping $f(s) = s \setminus \{ps\}$ maintains that for every transition $\langle x, l, y \rangle \in T$ from a reachable and relevant state $s$ with respect to the problems in $\mathcal{P}$, $\langle f(x), l, f(y) \rangle \in T'$. $\square$

### 5.2 Abstracting Preconditions

In the same scenario as above a user might want truck T1 to be used rather than truck T2. After restricting our problem to behave due to the contrast case our new plan would consist of truck T1 taking route 1, for the same reasons described in the previous chapter. As in the previous chapter this new plan $\pi$ takes 35 minutes compared to the original 25 minutes.

Through the use of abstractions we can determine the cause of the disparity in solution quality. In some cases abstracting away an entire predicate may be extreme and unnecessary. Instead we can abstract away certain preconditions for actions. If we abstract away the preconditions, $ps \in Ps$, that are responsible for checking that the produce is fresh before it is delivered, then we can produce a plan $\pi'$ in which truck T1 can perform route 2. This is a plan of the same quality as the original plan, but we are instead using truck T1 rather than truck T2 as the user expected. We can therefore produce an explanation such as "if the produce did not need to be fresh for it to be delivered, then truck T1 could be used such as in $\pi'$, otherwise our new plan will be $\pi$ which is slower by 10 minutes". An abstraction of a precondition consists of removing a predicate from all preconditions that the predicate appears in.

The formal process of abstracting a precondition from a planning model is as follows. Given a planning model $\Pi$ and a set of predicates $ps \subseteq Ps$ the abstracted model is $\Pi_{pre(ps)} = \langle D, Prob \rangle$, where $D = \langle Ps, Vs, As', arity \rangle$ and $As' = \{a' | \forall a \in As : Dur(a') = Dur(a), Pre(a') = Pre(a) \setminus \{ps\}, Eff(a') = Eff(a)\}$.

*Proof.* Removal of preconditions from actions is a valid abstraction. This is because removing preconditions manifests

as adding transitions in the LST, as can be seen from Definition 5 (*the ground action o represents is applicable in s*), so by Corollary 1, it is a valid abstraction.

By Corollary 1 this is an abstraction with $T' = T \cup \{\langle s \setminus \{ps\}, a, \mathit{Eff}_a(s \setminus \{ps\})\rangle)|\langle s, a, \mathit{Eff}_a(s)\rangle \in T\}$ where $\mathit{Eff}_a(s)$ is the result of updating $s$ with the effects of $a$. $\square$

## 5.3 Abstracting Durations

Again, assuming a user prefers truck T1 to be used rather than truck T2 in the example in Section 3. We would have new plan, $\pi$, that would consist of truck T1 taking route 1 and take 35 minutes compared to the original 25 minutes.

In our example, if we abstracted the duration of the action **drive_truck**, then we can produce a plan $\pi'$ in which truck T1 can perform route 1 quicker. An abstraction of an action's duration involves editing the action's duration constraint so that the planner can select any positive duration.

Given a planning model $\Pi$ and action schemas $as \subseteq As$ ,the abstracted model is $\Pi_{as} = \langle D, Prob \rangle$, where $D = \langle Ps, Vs, As', arity \rangle$ and $As' = As \setminus \{as\} \cup \{a'|\forall a \in as : Dur(a') = \langle 0, \inf\rangle, Pre(a') = Pre(a), \mathit{Eff}(a') = (a)\}$.

*Proof.* Our abstraction has no effect on the derived LST: $\tau == \tau'$, there is therefore no effect on the vertices and edges in the STN: $V' == V$ and $E' == E$, only $L' \neq L$.

For any two vertices $v_i, v_{i+1} \in V$ connected by an edge $e_1 \in E$ and $v_{i+1}, v_i$ connected by $e_2 \in E$, then $v_i, v_{i+1} \in V'$ and $e_1, e_2 \in E'$ and $l(e_1)' >= l(e_1)$ and $l(e_2)' <= l(e_2)$. This is because the labels in the STN are defined by the duration's of actions, the smallest possible duration of an action is 0, and the largest is inf. Therefore the labels in $G$ are contained within the labels of $G'$. $\square$

## 5.4 Abstracting Timed-Initial-Literals

Given the same scenario above, rather than abstracting the duration of the **drive_truck** action so that we can deliver the meat before it spoils, we could abstract the TIL that is responsible for the meat spoiling.

Through this abstraction we can produce a plan $\pi'$ in which truck T1 performs route 2. We can still deliver the meat when we arrive to location c in the unrefrigerated truck because it would not have spoiled. This is a plan of the same quality as the original plan, but we are satisfying the users foil posed in their question. We can therefore produce an explanation such as "if the meat did not spoil after 22 minutes, then truck T1 could be used such as in $\pi'$, otherwise our new plan will be $\pi$ which is slower by 10 minutes. An abstraction of a TIL consists of creating an action that models the TIL. This action can be performed at any time and will have the same effect as the TIL.

Given a planning model $\Pi$ and a set of TILs $t \subseteq T$ the abstracted model is $\Pi_t = \langle D, Prob \rangle$, where $D = \langle Ps', Vs, As', arity \rangle$ and $Ps' = Ps \cup \{p_t, d_t|\forall t\}$, and $As' = As \cup \{a_t|\forall t : Dur(a_t) = \langle 0, 0 \rangle, Pre(a_t) = \{p_t\}, \mathit{Eff}^x(a_t) = \{t_v\}, \mathit{Eff}^-(a_t) = \{p_t\}\}$, where $x$ is $+$ if the TIL is positive and $-$ otherwise; and $Prob = \langle Os, I, G \cup \{d_t|\forall t\}, M, T \setminus t$.

*Proof.* The abstraction of a TIL consists of modelling it as an action, which is exactly how TILs are handled in the LST and STN. This is an exact copy of the TIL in the LST and therefore the TIL is not changed and therefore $\tau == \tau'$. This differs in the LST only in that the action can now be executed at any time, i.e. $V' == V$ and $E' == E$, only $L' \neq L$. In the same way as the proof in Section 5.3 this is a valid abstraction as per Definition 10. $\square$

## 6 Searching for Suitable Abstractions

As described in Section 1 we consider the problem where we have a planning model, $\Pi$, with a solution, $\pi$, and a constrained model, $\Pi + c$, which we will call $\Pi'$, with a solution, $\pi'$, and there is a difference in quality between $\pi$ and $\pi'$.

We consider an abstraction $\alpha$ of a model $\Pi'$ to be suitable as part of an explanation if for both the solution to the abstracted constrained model $\Pi'_\alpha$, $\pi'_\alpha$, and the abstracted original model $\Pi_\alpha$, $\pi_\alpha$, have costs, $M(\pi'_\alpha)$ and $M(\pi_\alpha)$ and $|M(\pi'_\alpha) - M(\pi_\alpha)| < n$. Where n is a user-defined bound below which we consider plans to be *equi-cost*.

This creates a suitable explanation as it abstracts details from both of the models until they produce equi-cost plans. We can therefore say that it is those details that cause the discrepancy between the quality of the solutions for the original and constrained models. However, there may be many possible abstractions that produce equi-cost plans. To distinguish between these we are guided by the principle that we want to maintain, as much as possible, the most important and relevant structures of the problem. To this end we prioritise abstractions first by their size (fewer composed abstractions is better) and then by the similarity of the solutions to the abstracted and original problems, i.e the abstraction with the smallest $|M(\pi_\alpha) - M(\pi)|$.

### 6.1 Search

In this section we propose a search algorithm to identify correct abstractions that produce informative explanations.

The approach we take is to search over a poset of possible abstractions. Definition 12 allows us to determine a poset given a planning model, $\Pi$, the set of possible abstractions, $\alpha$, as $L(\Pi, \alpha) = (\mathcal{M}_\Pi, \leq)$. This gives an ordering on the models that can be reached through the set of abstractions $\alpha$. This underpins our approach of abstraction for explanation.

**Definition 12.** *Given a planning model,* $\Pi$*, and a set of abstractions,* $\alpha$*, the collection of* associated *models,* $\mathcal{M}_\Pi$*, is the closure of the image of* $\Pi$ *under the application of the abstractions,* $\alpha$*. The partial ordering on* $\mathcal{M}_\Pi$ *is defined by: for* $\Pi_1, \Pi_2 \in \mathcal{M}_\Pi$*,* $\Pi_1 \leq \Pi_2$ *iff* $\rho(\Pi_1) \subseteq \rho(\Pi_2)$*. This determines the poset* $L(\Pi, \alpha) = (\mathcal{M}_\Pi, \leq)$*.*

For an abstraction poset, elements higher up in the poset are more abstract than those at the bottom. The abstractions that will form $\mathbb{A}$ in our search are all the possible abstractions defined in Section 5. We do not construct this poset a priori, we generate parts of the poset during search time.

We provide a search algorithm for finding suitable abstractions for explanations in Algorithm 1. This algorithm implements a bottom-up breadth first search. We start from the constrained model and apply all possible abstractions

at each level of the poset before traversing up the poset in a single step. This is repeated until we reach a model that produces an equi-cost plan to the original solution. This algorithm take as input the constrained model, $\Pi'$, the set of possible abstractions, $\mathbb{A}$, (see Sec. 5), the original plan, $\pi$, and a real value specifying the bound for equi-cost plans, $n$. A queue is constructed, $queue\mathbb{A}$, from the abstractions, $\mathbb{A}$, the current abstraction, $curr_\mathbb{O}$, is then dequeued. The first model that abstractions are applied to, $curr_\Pi$, is a copy of the model $\Pi'$. This is abstracted with $curr_\mathbb{O}$ to give the abstracted model, $\Pi'_\mathbb{O}$, the function $abstract$ takes an abstraction, $\mathbb{O}$, and a model, $\Pi$, and returns an abstracted model, $\Pi_\mathbb{O}$. A queue, $queue\Pi$, is constructed from the abstracted model, $curr_\mathbb{O}$, that model is then solved, the function, $solve$, takes a model, $\Pi$, and returns a valid solution plan for, $\Pi$. The algorithm then checks if the abstracted plan and original plan are equi-cost, the function $bound$ is as described in the introduction to this section. The algorithm then loops, repeating this process for each model in the queue, for each abstraction in the queue until there are no models left in the queue. In each iteration there is a further check that an abstraction, $curr_\mathbb{O}$, has not already been applied to the current model, $curr_\Pi$.

Algorithm 1 gives a satisficing solution. It gives the first of abstractions that produces a model whose solution is equi-cost to the original. If we assume that all abstractions are of equal cost, then this would be an optimal solution as we will apply all possible abstractions at each level of the poset before moving on. However, as noted earlier in this section, we can evaluate abstractions based on the distance between the quality of the solution of the abstracted model and the original solution. We can modify the stopping condition of Algorithm 1 to give an optimising search algorithm that evaluates the quality of certain solutions based on this metric.

### 6.2 The Explanation

The explanation will be composed of the abstractions that were used in order to get a satisfiable plan under the restricted model. The specific explanation will be contextualised by the constraint applied to the original model that lead to a difference in the quality of the solutions. Each abstraction $\mathbb{O}_{curr}$ is popped from the set of abstractions $\mathbb{O}$. The type of the abstraction is then checked, these are each of the abstractions outlined in Section 5. The explanation is then formed by the types of, and the abstractions that allow the problem to produce an equi-cost solution. In the next section we give some examples of these generated explanations.

## 7  Experiments and Results

In this section we present an empirical evaluation of our approach to finding explanations. Whilst we present results indicating the performance of our algorithm; our focus is on testing the ability of our approach to find explanations. Optimizing the efficiency of the algorithm is left to future work.

Evaluation uses 6 differently structured planning problems. 2 of these problems are based on our running example and the other 4 are instance 1 of Rovers and 2 of Satellite from the International Planning Competition (IPC) (Long and Fox 2003). The 2 explainable problems are the Delivery

---

**Algorithm 1:** Breadth first search over an abstraction poset $L(\Pi', \mathbb{A})$ to find a set of abstractions $\mathbb{O} \in \mathbb{A}$ supporting a plan $\pi'$ such that $|M(\pi') - M(\pi)| < n$

**Data:** $\{\Pi, \Pi', \mathbb{A}, n\}$
**Result:** $\mathbb{O} \subseteq \mathbb{A} \vee FAIL$

1   $queue\mathbb{A} \leftarrow queue(\mathbb{A})$;
2   $curr_\mathbb{O} \leftarrow dequeue(queue\mathbb{A})$;
3   $\Pi_\mathbb{O} \leftarrow abstract(\Pi, curr_\mathbb{O})$;
4   $\Pi'_\mathbb{O} \leftarrow abstract(\Pi', curr_\mathbb{O})$;
5   $queue\Pi \leftarrow queue(\Pi_\mathbb{O})$;
6   $queue\Pi' \leftarrow queue(\Pi'_\mathbb{O})$;
7   $\pi \leftarrow solve(\Pi_\mathbb{O})$;
8   $\pi' \leftarrow solve(\Pi'_\mathbb{O})$;
9   **if** $|M(\pi') - M(\pi)| < n$ **then return** $\mathbb{O}$ ;
10 **while** $\neg empty(queue\Pi')$ **do**
11    **while** $\neg empty(queue\mathbb{A})$ **do**
12     $curr_\mathbb{O} \leftarrow dequeue(queue\mathbb{A})'$;
13     **if** $\neg applied(\Pi'_\mathbb{O}, curr_\mathbb{O})$ **then**
14      $\Pi_\mathbb{O} \leftarrow abstract(curr_\Pi, curr_\mathbb{O})$;
15      $\Pi'_\mathbb{O} \leftarrow abstract(curr_{\Pi'}, curr_\mathbb{O})$;
16      $enqueue(queue\Pi, \Pi_\mathbb{O})$;
17      $enqueue(queue\Pi', \Pi'_\mathbb{O})$;
18      $\pi \leftarrow solve(\Pi_\mathbb{O})$;
19      $\pi' \leftarrow solve(\Pi'_\mathbb{O})$;
20      **if** $|M(\pi') - M(\pi)| < n$ **then return** $\mathbb{O}$ ;
21    $\Pi_\mathbb{O} \leftarrow dequeue(queue\Pi)$;
22    $\Pi'_\mathbb{O} \leftarrow dequeue(queue\Pi')$;
23    $queue\mathbb{A} \leftarrow queue(\mathbb{A})$;
24 **return** $FAIL$;

---

domain discussed in Section 3 and an augmented version of this domain in which there is another driver, D1, and only one driver, D2, has the ability to drive the refrigerated truck, we call this problem **delivery+**.

In Rovers, rovers take samples of soil, rock, and image data to send back to a lander. Not all rovers are equipped for all sample types. Cameras must be calibrated to take images, rock data must be stored before it can be communicated, and soil data can be collected and communicated. The optimal plan for problem 1 is to use rover R1, calibrate the camera and take an image, traverse to and collect the rock sample, store it, and communicate the data; and finally traverse to the soil sample, collect it, and communicate the data.

In our Satellite problem there is a satellite with two instruments able to take photos in different modes. Instrument I1 can take infrared images and I2 can take both infrared and visible images. The goal of the problem is to take images of certain targets in certain modes. The optimal plan is to turn on and calibrate I2 and take the required images.

Each problem is solved, then constraints added to make the problem unsolvable or to require a plan of worse quality than the original. We search for suitable abstractions using the optimising version of Algorithm 1. Experiments ran on a 4gb machine with an i7-12800H CPU, and a 4 hour timeout. We use the full set of abstractions detailed in Section 5.

The delivery problem is constrained so that truck T1 must

| Domain | Size of $\mathbb{A}$ | First/s | Optimal/s | # Suitable | # Optimal | Size | # Nodes | Time/s |
|---|---|---|---|---|---|---|---|---|
| Delivery | 21 | 20.04 | 20.04 | 489 | 203 | 1 | 585 | 20 |
| Delivery+ | 25 | 220.37 | 1382.22 | 307 | 12 | 2 | 868 | 20 |
| Rovers Worse | 64 | 520.61 | 1802.31 | 33 | 2 | 1 | 178 | 40 |
| Rovers Unsolvable | 64 | 10295.09 | 10295.09 | 10 | 4 | 2 | 350 | 40 |
| Satellite Worse | 22 | 420.31 | 420.31 | 11 | 8 | 1 | 52 | 140 |
| Satellite Unsolvable | 22 | 280.20 | 280.20 | 26 | 16 | 1 | 79 | 140 |

Table 1: Time taken to find and number of suitable and optimal abstractions for a selection of planning problems.

be used, requiring the poorer route 1 to be taken. The delivery+ problem is similarly constrained, but also so that driver, D1, who cannot use the refrigerated truck, must make the deliveries. This is unsolvable.

The Rovers problem is constrained so that the image must be taken before the camera is calibrated. The resulting plan has the image capture in a different place in the solution. The problem is separately constrained by removing the capability of the rover to sample soil or rocks, which is unsolvable.

The Satellite problem is constrained so that instrument I1 must be used to take an image; the plan then uses I1 to take the infrared images and instrument I2 is switched on to take the visible light images. The problem is separately constrained so I2 cannot be used, which is unsolvable.

The results in Table 1 show, for each problem, the number of abstractions at each search level, (the size of $\mathbb{A}$); the time to find the first suitable abstraction (henceforth, abbreviated to *fsa*); the time to find the first optimal abstraction; the numbers of suitable and of optimal abstraction sets for explanations found; the size of the abstraction set; the number of nodes expanded in the search; and finally the time bound given for each problem. The bound is calculated by finding the largest time taken to solve the original problem, or any of the constrained problems, and doubling it.

The *fsa* for the delivery problem was found in 20.04 seconds; it is also optimal. 585 combinations of abstractions were searched of which 489 were suitable and 203 optimal. The first optimal solution found, of size 1, abstracts away the predicate determining whether meat or cereal is in a truck. Now, the action that prolongs the life of the meat can be used with truck T2 as the meat is no longer required to be in the truck for this action to be performed. The plan is then to use truck T1 to perform route 2 whilst using truck T2 to act as the refrigerator from location loc_a. The explanation produced is: {*Predicate: (in ?p - produce ?t - truck)*}. There are many optimal solutions for this problem. Another is to abstract the TIL responsible for determining when the meat spoils. The plan for this abstracted problem uses truck T1 to perform route 2 without worrying about the meat spoiling at all. The explanation is: {*TIL: (fresh ?m - meat)*}.

The *fsa* for the delivery+ domain is found in 220.37 seconds. The first optimal abstraction is found in 1382.22 seconds. 868 combinations of abstractions were searched. of which 307 were suitable and 12 were optimal. The first optimal solution found, of size 2, abstracts away both the predicate for a truck to be refrigerated and the requirement for a driver to be qualified to drive certain trucks. Then truck T1 can be used in place of the refrigerated truck, and any driver

can drive any truck. The plan for this abstracted problem is to use T1 with the driver D1 to perform route 2.

The *fsa* for the rovers problem constrained to produce a worse quality plan was found in 520.61 seconds. The first optimal abstraction was found in 1802.31 seconds. 178 combinations of abstractions were searched, of which 33 were suitable and 2 optimal. The first optimal solution found abstracts away the predicate responsible for ensuring the target of calibration matches is the objective image to be taken. This allows for a camera to be calibrated on any target and so the original calibration action never has to be performed and so the constraint is trivially satisfied. The plan continues the same as in the original plan, after the camera is calibrated with an arbitrary objective.

The *fsa* for the rovers problem constrained to be unsolvable is found in 10295.09 seconds; it is also optimal. 350 combinations of abstractions were searched, of which 10 were suitable and 4 optimal. The first optimal solution, of size 2, abstracts away the preconditions of the **sample_rock** and **sample_soil** actions that ensure the rover has the ability to sample rock and soil; any rover can then gather rock and soil samples and the plan is then the same as the original. The explanation is: {*Precondition: (equipped_for_soil_analysis ?r - rover),(equipped_for_rock_analysis ?r - rover)*}

The *fsa* for the Satellite problem constrained to produce a worse quality plan is 420.31 seconds; it is also optimal. 52 combinations of abstractions were searched, of which 11 were suitable and 8 optimal. The first optimal solution, of size 1, abstracts the precondition of the **take_image** action responsible for ensuring the instrument taking the image supports the relevant mode. The instrument I0 can then be used to take images of visible light and the plan for the abstracted model is the same as the original plan other than in using instrument I0 instead of I1.

The *fsa* for the Satellite problem constrained to become unsolvable is 280.20 seconds; it is also optimal. 52 combinations of abstractions were searched of which 26 were suitable and 16 optimal. The first optimal abstraction and resulting plan is the same as the other Satellite problem.

## 8 Conclusion

In this paper a general definition for abstractions in planning problems is presented and a number of surgeries on planning models are defined. A proof-of-concept algorithm is presented and used to find suitable and optimal abstractions. Building natural language explanations from these abstractions and evaluating them remains future work.

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
