# OpenReview forum: "Explaining Plan Quality Differences"
_icaps-conference.org/ICAPS/2024/Conference — ICAPS 2024_

### Official Review · Reviewer_4VZF · 2024-01-18

**Significance And Importance:** 2
**Soundness:** 3
**Novelty:** 3
**Clarity:** 3
**Overall Evaluation:** 2
**Confidence:** 4

**Weaknesses:**

0: Minor weaknesses requiring some work to be addressed for the paper to be accepted.

**Contributions Of The Paper:**

The paper provides an approach to explaining why two related models generate plans with different plan quality. This can be used as a supporting system for an XAIP-as-a-Service approach, which creates contrastive plans, but does not explain why these plans have better/worse quality. In this paper model abstractions are considered as the options for generating explanations. The approach starts with the original and the constrained models. Model abstractions are then applied until the two models generate plans of equivalent quality. The expectation is that if an abstraction leads to the plans having the same metric score then the abstraction provides an explanation for the observed difference in plan metric. In the evaluation they examine some specific scenarios.

**Ethical Considerations:**

(1) Not Applicable: The paper does not have any ethical considerations to address

**Nomination For Best Paper:**

No

**Questions For Authors:**

1. Can you report on the fraction of optimal explanations that correspond to the actual change that was made? How does this change with more modifications?

2. Is there a scenario, where providing an explanation after 30 minutes is useful?

3. Did you explore the impact of parameter n? I don't see any reason for this not to be 0. I understand that with a value higher than 0 you might allow for smaller abstractions that miss details that only account for marginal difference. However, this is at the cost of weakening the only criteria you have for pruning out incorrect explanations. From my understanding you will increase the set of satisfying candidate abstractions. Is there any evidence that this parameter can lead to a reasonable compromise?

---
Thanks for your responses.

**Reproducibility:**

3: Authors describe the implementation and domains in sufficient detail.

**Strengths Of The Paper:**

The motivation of this work is good. Trying to generate an explanation to support XAIP-as-a-Service explanations is very interesting and relevant. The use of abstractions has been considered in related work and been shown to be effective. The paper spends effort laying the framework for the abstractions in this work. The selected abstractions seem to cover appropriate aspects, and I appreciated the inclusion of abstractions for temporal models. A general search strategy is outlined for discovering sets of abstractions that explain the difference in plan quality.

The presented approach is promising and can make a useful contribution to the community.

**Weaknesses Of The Paper:**

An issue here is the granularity of the abstractions. I appreciate that it is not the point here to consider the generation of text explanations, however, given that the explanation is at the level of predicate/action duration etc., I am concerned that the generated explanations lack the necessary information content to make useful text explanations. E.g., compare the explanations in the paper  for the specific TIL against the explanation for removing a predicate from the domain.

The actual approach was surprising to me. There is no real discussion about what we can expect when we add a constraint to a model. My view would be that it only makes sense to use an explanation approach of plan quality if the setup is expected to generate plans of consistent quality (e.g., that the time-limit is suitable for the problem). And in this case we can expect that the new plan will have the same or worse quality. In the case of the same quality, there is nothing to explain. Otherwise, it seems that the simulation of the original plan in the constrained model would provide a starting point for an explanation. This would potentially allow for more guidance, which might allow for more specific abstractions.

The main limitation for me is that the evaluation does not properly investigate the approach, and didn't provide any insight into when the approach would be useful. Of course there are multiple explanations generatable, but how would this be used in practice? The paper suggests that a useful explanation might be one of those generated by the approach, but it is impossible to understand how many of the generated explanations are useful and how this would degrade as more modifications were made. I think that the evaluation needs motivated with a scenario where this approach, with its associated runtime, can be justified.

Notes:
In XAIP-as-a-service I would expect the structure for the constraints to be automatically generated. I appreciate that you do not aim to generate text explanations here, but this seems to be a problem.

The properties of the approach are not really discussed.

The parameter of n used in the evaluation is (I believe) missing from the paper.

---

> ### Author Rebuttal · Authors · 2024-01-26
>
> Thanks for your detailed review.
>
> 1) Whether an explanation corresponds to the constraint added is not always an indication of a good explanation. E.g., in the refrigeration domain the constraint is to use truck 2; but the explanation needs to be in terms of refrigeration or lifespan of the meat rather than that truck 2 cannot be used. Perhaps the spirit of the question is more: what proportion of optimal explanations are `good’ ones? See answer to point 6.
>
> 2) See R: bkfE point 4 for a scenario.
>
> 3) See R: AkR9 point 6.
>
> Weaknesses:
> Labeling paragraphs 4, 5, and 6.
>
> 4) The generation of natural language explanations is future work. However using information from the action models, problem instance, and plans, we believe we can derive explanations like the Section 5 examples.
>
> 5) A constraint has been added to the problem: if this does not render the original plan invalid there is nothing to explain. Otherwise, simulating the plan is often not useful: for example, in our refrigeration domain if the constraint is “use truck 2”, then VAL will report that the goal (using truck 2) is not satisfied, which we already knew. If the constraint is “don’t use truck 1”, then VAL will report the original plan invalid because it uses truck 1. We have thought about this and how more sophisticated analysis (e.g. finding landmarks) might help when we consider guiding search because we want to present a general approach, not relying on the domains/constraints having specific characteristics.
>
> 6) An optimal abstraction for an explanation minimizes plan quality difference. The number of explanation abstractions for each domain is in Table 1. Barring the delivery domain (which, with restriction to the optimal solutions of the smallest size, we reduce to 8), the number of solutions is small. The explanations reported in the evaluation are simply the first solutions found: these are promisingly informative. Future work, including user studies, is planned to identify the best metrics to determine which abstractions make good explanations. This is not straightforward, as different explanations might better serve different users. There might be some abstractions that are actionable (eg one could put 5 extra litres of fuel into their car to make it home), whereas some are not actionable, (eg move a building out of a path).
>
> We search over combinations of abstractions and report those that are optimal by our metric. For a justification of runtime see R: bkfE point 4 for a scenario.

---

### Official Review · Reviewer_bkfE · 2024-01-20

**Significance And Importance:** 2
**Soundness:** 3
**Novelty:** 3
**Clarity:** 4
**Overall Evaluation:** 2
**Confidence:** 3

**Weaknesses:**

1: Minor weaknesses that are easily fixable.

**Contributions Of The Paper:**

The paper focuses on explaining the difference between quality solutions in automated planning, for this, the authors propose an implementation of some abstractions to explain the differences in temporal problems. They propose to create a graph where the vertices are the points and the wight edges present the max/min separation between points to make this abstraction. Later, this paper presents a successful experimentation with a reduced set of problems.

**Ethical Considerations:**

(4) Good: The paper adequately addresses most, but not all, of the applicable ethical considerations

**Nomination For Best Paper:**

Yes

**Questions For Authors:**

Do you think this technique, regardless of temporal domains, is generalizable?
Could this abstraction be used in numerical domains?

**Reproducibility:**

3: Authors describe the implementation and domains in sufficient detail.

**Strengths Of The Paper:**

The abstraction for temporal-numeric planning domains is not well-done research and this paper presents a theoretically well-explained approach. The authors define a valid abstraction of a planning model as based on their derived LST and STN.
The experimentation is done for solved and Unsolvable problems.
The paper is very well written and clear, and I think this research is great for this conference.

**Weaknesses Of The Paper:**

The used domains are pretty similar, and I don’t know if this abstraction only is good for these three domains. I would like to know what intuition the authors have in which areas this might not work well.

I don’t understand why 4 hours for experimentations only rovers unsolvable need more than 600 second to find first solution. This time a satellite or rover would involve a high battery cost and would not be applicable in real environments.
The first solution in delivery problem is the optimal (this domain in simple) and it takes around 20 second to exact it, I don’t really need run 4 hours for that or to find another solution. I think the limitation of the experimentation should be 3 hours for these domains. At least the authors  know that this technique only resolves in less than 3 hours for these domains and in others nothing is achieved.

---

> ### Author Rebuttal · Authors · 2024-01-26
>
> Thanks for your detailed review.
>
> 1) This approach can be used to provide explanations for quality differences in classical/temporal/numeric planning problems (for numeric see below). If there is a feature of a planning language that is not representable in these languages, then definitions of validity can be used as tools to generalize this approach to new abstractions that are useful for these planning problems. The majority of planning problems can be represented with an LST and STN, and so these definitions of validity will still apply. Other models of abstraction could be implemented using these tools.
>
> 2) We have an implementation for abstracting non-continuous numeric functions for providing explanations in numeric domains, but decided to focus this paper on the proposition temporal case.
>
> Weaknesses:
>
> 3) “A weakness is evaluating the approach on similar domains, and choosing a timeout of 4 hours”: Our focus here is on presenting a framework for using abstractions for explaining plan quality differences. Since we are not yet claiming to have efficient search, we instead ran this experiment to prove that this process can give us abstractions that are suitable for explanations. We deliberately chose varied domains in which there was inherent explanatory content (refrigeration) and unknown explanatory content (rovers and satellite). The 4 hour timeout was chosen to give us a better characteristic of this process in terms of the number of suitable and optimal abstractions that exist within this search space. We also have some additional results on other domains (goldminer, crewplanning, depots and elevators) but were restricted by space. We also believe that a large scale evaluation is not feasible until we have an optimized algorithm. Improving the runtime for this search is the second step to this process that we are currently working on.
>
> 4) “Using 600 seconds to find an explanation in satellite/rovers”: We do not suppose that one would run the planner on a rover or a satellite. The explanation process would be used by ground teams making plans for rovers/satellites. Planning takes place over the course of several days for a day of operations, and, while the faster an explanation the better, waiting 10-30 minutes for an explanation is not unreasonable. Our explanations are currently aimed at humans.

---

### Official Review · Reviewer_AkR9 · 2024-01-22

**Significance And Importance:** 2
**Soundness:** 3
**Novelty:** 2
**Clarity:** 3
**Overall Evaluation:** 1
**Confidence:** 3

**Weaknesses:**

0: Minor weaknesses requiring some work to be addressed for the paper to be accepted.

**Contributions Of The Paper:**

This paper introduces a novel methodology for explaining the difference in quality between two plans in a planning model. This is achieved by abstracting away elements from the planning model until the quality metrics of the two plans match. The abstraction process involves simplifying the problem by removing details from the model with the lesser quality plan, thus preserving valid plans and making the problem easier to solve. This approach extends existing work on abstraction in planning, primarily used for computing heuristics, to a new application of explaining plan quality differences. The methodology defines the space of legal abstractions and details the process of creating these abstractions, representing a significant advancement in the field of automated planning by providing a systematic way to understand and explain plan quality variations.

**Ethical Considerations:**

(1) Not Applicable: The paper does not have any ethical considerations to address

**Nomination For Best Paper:**

No

**Questions For Authors:**

1) How do the authors propose to manage the inherent complexity of the abstraction process, especially for users who may not be deeply familiar with the intricacies of planning models and abstractions?
2) What measures are in place to ensure the accuracy of the model abstractions? How do the authors address the potential impact of inaccuracies in abstractions on the explanations provided?
3) Can the authors elaborate on the effectiveness of their methodology in extremely complex planning domains? Are there limitations to the abstraction process in scenarios with numerous interacting elements and constraints?
4) How does the methodology account for computational resource limitations, especially in large and complex planning models? Are there any strategies to improve the efficiency of the abstraction process?
5) How generalizable is the methodology across different planning problems? Can the authors discuss its applicability and limitations in various types of planning domains?
6) What is the impact of user-defined parameters and the inherent subjectivity in the process? How can different users ensure consistency in the explanations for the same plan quality differences?
7) How do the authors address the risk of oversimplification in the explanations, especially when multiple factors contribute to plan quality differences?
8) Can the authors provide more insights into the scope of their evaluation? Are there plans to extend the evaluation to cover a broader range of scenarios and planning problem variations?

**Reproducibility:**

2: Some details are missing, but the paper still appears to be replicable with some effort.

**Strengths Of The Paper:**

-- Systematic Abstraction Process: The paper systematically demonstrates how to abstract various components of a planning model, such as predicates, preconditions, and action durations. This comprehensive approach allows for a detailed examination of how different aspects of a model contribute to the overall plan quality.
-- Valid Abstractions Guarantee: The methodology ensures that the abstractions made are valid, meaning they always preserve the feasibility of the plans. This is crucial for ensuring that the explanations provided are based on realistic modifications of the planning model.
-- Empirical Evaluation: The paper includes an empirical evaluation of the approach, applying the methodology to different structured planning problems. This not only demonstrates the applicability of the approach but also provides insights into its effectiveness.
-- Search Algorithm for Suitable Abstractions: The introduction of a search algorithm to identify suitable abstractions for explanations adds a practical dimension to the theory. This algorithm helps in finding the most relevant abstractions for a given planning problem.
-- Focus on Explanations for Plan Quality: The paper addresses a gap in the literature by focusing on explaining plan quality differences, a topic that is crucial for the practical application of planning models but not often addressed in depth.

**Weaknesses Of The Paper:**

-- Complexity of Abstraction Process: The process of abstracting planning models to explain plan quality differences is inherently complex. This complexity might make it challenging for users without a deep understanding of planning models and abstractions.
-- Dependence on Accurate Model Abstractions: The effectiveness of the methodology heavily relies on accurately abstracting the planning models. Any inaccuracies in abstraction might lead to incorrect or misleading explanations.
-- Limitations in Handling Highly Complex Domains: While the approach is novel, its effectiveness in extremely complex planning domains with numerous interacting elements and constraints is not fully explored. The abstraction process could become overly complicated or less effective in such scenarios.
-- Computational Resources and Efficiency: The process of finding suitable abstractions and generating explanations might require significant computational resources, especially for large and complex planning models. This could limit its applicability in resource-constrained environments.
-- Generalizability of the Methodology: The paper's methodology, while innovative, might have limitations in its applicability to a wide range of planning problems. The specific types of abstractions and their implications for different planning domains are not extensively discussed.
-- User-Defined Parameters and Subjectivity: The process involves user-defined parameters (like the threshold for equi-cost plans), which introduces a degree of subjectivity. Different users might arrive at different explanations for the same plan quality differences.
-- Potential Over-Simplification in Explanations: There is a risk that the explanations generated might oversimplify the actual reasons behind plan quality differences, especially in cases where multiple factors contribute to the disparity.

---

> ### Author Rebuttal · Authors · 2024-01-26
>
> Thanks for your detailed review.
>
> 1) The user does not deal with the abstractions, they are automatically generated and searched by the system. The user does not have to suggest abstractions or generate the abstractions. They only have to understand the meaning of the explanation. The current form of our explanation is to specify what has to be done to the domain (e.g. remove the refrigerated predicate).
>
> 2) First, we assume that the planning model is correct, and we prove that each abstraction is sound in the paper. The question is whether we find the ‘right’ abstraction, to do this we search over abstractions and provide the best one according to our metric; of course alternatively we can provide several possible explanations for the user if there are multiple suitable abstractions. In addition, explanation is often an iterative process, where the user provides additional constraints that refine the possible answers and explanations at each step.
>
> 3) We propose a set of abstractions that we have recognised as useful for explanation. We search over this set of abstractions, at each level of the search tree applying depth + 1 abstractions. In more simple domains a single abstraction may be suitable for an explanation. However, in more complex domains it may be necessary to use a combination of abstractions as the basis of the explanation. For example, in the Delivery+ and Rovers Unsolvable domains the optimal explanation consisted of two abstractions.
>
> 4) See R: bkfE point 3.
>
> 5) See R: bkfE point 1.
>
> 6) Some constraint has been added to the planning problem and so the model must produce a different plan. For that different plan to have the exact same quality as the original plan leaves very little wiggle room. For example, in the refrigeration domain the constraint may be to take a slightly longer path, that might be a minor difference and the major difference is meat spoiling. In our evaluation we used n=2, but this is a parameter that can be adjusted. An example of when this might be practically useful is in a delivery domain, a CEO may know that there is an absolute limit on cost and so ensure it does not change, whereas an engineer might want to understand why certain routes are worse in spite of that limit. Determining the best value is future work. There is only one user defined parameter. If user’s need to preserve the consistency of explanations, they can publish the parameter they chose.
>
> 7) This is answered by point 3.
>
> 8) See R: bkfE point 3.

---

### Meta-Review · Area_Chair_ndJJ · 2024-02-05

**Recommendation:** Accept (Oral)
**Confidence:** 4

**Metareview:**

This is a short metareview: everybody liked the paper, and no further discussion was needed to come up with an "accept" recommendation. Congratulation on a nice paper, looking forward to seeing it at ICAPS!

**Ethical Considerations:**

(1) Not Applicable: The paper does not have any ethical considerations to address